# OpenReview forum: "Injecting Falsehoods: Adversarial Man-in-the-Middle Attacks Undermining Factual Recall in LLMs"
_TMLR — Rejected by TMLR_

### Review · Reviewer_HD7x · 2026-02-23

**Summary Of Contributions:**

This paper explores the vulnerability of LLMs under a man-in-the-middle (MitM) setting where prompts are adversarially modified during transmission. The authors propose three types of prompt manipulation strategies intended to mislead the model into producing incorrect answers, and report high empirical attack success rates. In addition, a defense mechanism based on response uncertainty is introduced.

**Audience:**

Yes

**Audience Explanation:**

1. The vulnerability of LLMs against adversarially attack is a timely and important topic.

2. The effort to provide a mathematical formulation of the problem is appreciated, although the soundness and practical relevance of the proposed definitions could be further strengthened.

**Claims And Evidence:**

No

**Claims Explanation:**

1. Practical relevance of the threat model. While the paper aims to demonstrate that LLMs are vulnerable to adversarially perturbed prompts, the practical significance of the MitM setting is not yet fully justified. The motivation would be substantially strengthened by concrete real-world scenarios/examples in which such attacks could realistically occur. Moreover, if an attacker is able to intercept and modify the prompt during transmission, it seems that they might also be able to directly manipulate the returned response. Clarifying why the prompt-level attack is the primary or more realistic threat in this setting would help better position the contribution.

2. Overstatement of contributions. The paper uses phrases such as “first principled attack evaluation on LLM factual memory” and “theory-grounded MitM framework.” At present, these claims appear stronger than what is technically supported: defining three abstract attack types does not yet substitute a framework, and the paper does not provide theoretical analysis of the proposed attacks or defenses.

In addition, several modeling choices raise concerns regarding realism and rigor:

   i. In section 3, (1) The function $g$ is modeled as deterministic, whereas LLM outputs are inherently stochastic. (2) in practice, a query often corresponds to multiple correct answers, e.g., date or name formats, which is not considered in the formulation.

   ii. Regarding Definition 1, an attack does not need to flip the answer for every query in practice. Moreover, once the output is probabilistic, it becomes unclear how the attack success condition should be formally defined.

   iii. For definition 2-4, it is unclear how to solve optimization objectives (2-4) in practice. These objectives require access to $g$, which an attacker typically does not have.
   Moreover, fact-aware attacks assume the availability of a ground-truth fact extraction function $w$ and other auxiliary functions such as $h$ and $\Omega$. These are strong assumptions and their existence or accessibility in realistic scenarios is unclear.


   Overall, the current definitions appear to be attempts capturing the ideas behind three attacks proposed by authors, rather than well-justified, operational formulations. Some assumptions are quite idealized, and the absence of theoretical analysis makes it difficult to view the approach as “theory-grounded.”

**Requested Changes:**

Besides the two main concerns regarding practical relevance and contributions, I have the following concerns:

3. Questions regarding experiments.

​	i. The authors acknowledge that they construct their own datasets to implement $\beta$- and $\gamma$- type attacks, which reinforces my concern that such attacks may be infeasible in practice, as attackers typically have limited knowledge of the underlying data and application.

​	ii. In Figure 5, the meaning of “uncertainty between correct and incorrect answers” is unclear. Why is the absolute difference used instead of the signed difference?

​	iii. The attack success rate is computed as one minus accuracy. However, a more natural definition would count only cases where the model’s output changes from correct (without attack) to incorrect (under attack). Errors that occur even without an attack should not be considered successful attacks. Reporting ASR under this definition would be informative.

​	iv. While GPT-4o achieves high accuracy without attack, it also has pretty high accuracy under $\alpha$- type attack, which explicitly instructs the model to gives wrong answers. Can authors elaborate the rationale behind?

​	v. Since LLMs can produce incorrect answers with high confidence (i.e., low uncertainty), it would be useful to further justify why uncertainty is expected to be a reliable signal for attack detection.

4. The writing, especially presentation of several mathematical statements, can be improved for clarity and precision. A few examples:

   i. In Section 3, the statement $\Phi(q_i,a_i)=1 \forall i \ s.t. q_i \in \mathcal{Q}, a_i\in\mathcal{A}$ is difficult to interpret. It seems the intended meaning is: "For every $q_i\in \mathcal{Q}$, there exist an answer $a(q_i) \in\mathcal{A}$ such that $\Phi(q_i,a(q_i))=1$, and it equals zero for other answers."

   ii. In Eq (1), the argument of the argmin is unclear. Also, what is $q_i^*$? Similar issues appear in Eqs. (2–4), e.g., in Eq (2), why the left hand side is $a_i^\star$? There is also a typo after Eq. (3): $f_j=w(q_i)$ instead of $f_j=z(q_i)$.

   iii. The definition and motivation of $h$ are currently difficult to follow.

   iv. In Figure 4, ordering models by size would make the comparison easier to interpret.

​	Overall, I would suggest the authors to throughly review and revise the manuscript.

---

> ### Author Response · Authors · 2026-03-17
>
> We thank the reviewer for the thorough evaluation of our work and the provided feedback. Below you find the answers to the concerns you raised, which we have addressed in the updated manuscript. *Hence, please also refer to the uploaded revised manuscript, which includes the changes marked in blue, and deletions marked in red.*
>
> **1) Threat model:**  Thank you for raising this point - we believe the manuscript would indeed benefit from the following clarification. We focus on query-level perturbation due to the asymmetric integrity protection prevalent in enterprise LLM deployments. While a MitM adversary may be able to access traffic in both directions, modern API gateways typically implement Unidirectional Response Signing (e.g. via HMAC) to ensure output integrity. In this architecture, the gateway verifies that the response originates from a trusted LLM IP address and matches a specific cryptographic signature; any downstream modification by an attacker would invalidate this signature and trigger a security rejection. From a scalability and impact perspective, we believe that a query-side attack is more parsimonious and more scalable as the attacker could exploit specific query types on certain topics potentially affecting millions of users at the same time. Very concrete real-world examples may include scenarios like safety disinformation (e.g. intercepting a consumer's inquiry regarding a product's safety recall to report a hazardous item as safe), reputational sabotage (e.g. intercepting a query about a public official's information to inject a false claim of a criminal investigation), or similar imaginable scenarios.
>
> **2) Overstatement of contributions:** We revised our contributions accordingly to loosen our claims. Moreover, we have significantly reworked the formalizations in Section 3 and 4 to further simplify them.
>
> **i)** (1) Indeed, you are correct. We have relaxed the definition of function $g$. (2) We have added a footnote explaining this.
>
> **ii) and iii)** We have significantly simplified the definitions in the paper. These concerns should hence be resolved now.
>
> **3) Experiments:**
>
> **i)** As for the fact-aware attacks, beta and gamma, we assume the attacker has access to a fact-checker function. In reality, this would be easily constituted by a short lookup in a search engine of choice (in Section 4.2 constituted as  $\Omega$, an external knowledge base containing facts). If the attacker then knows what the correct answer is, they have unlimited possibilities of an incorrect fact (or an unrelated fact, as with gamma) to append to the original query. We therefore believe the feasibility of this setup in reality is given.
>
> **ii)** We use the signed difference because of the nature of our metrics (entropy, perplexity, token probability). While with the first two, a lower value indicates lower uncertainty, the opposite holds for token probability (higher value = less uncertainty = better). Therefore, we report the absolute differences simply to avoid confusion. This is true of all models: attacked answers have higher uncertainty. But expressing it in signed values would show positive amounts for entropy and perplexity, and negative amounts for token probability, even though the results say the same thing. Hence, the absolute values simply show a noticeable difference in uncertainty between the attack and no-attack conditions.
>
> **iii)** In fact, this is what we do. As stated in Section 5.1, “Baseline performance,” we identify the questions the models can answer correctly without any attack. With these questions, we then later conduct the attack experiments. In other terms, the ASR does not include cases of wrong answers in non-attack settings.
>
> **iv)** This is a good point. It might be that GPT-4o (as well as Mistral, for that matter: both the old Mistral-7B as well as the newly added Mistral-Nemo perform well under alpha attack) is more stable against instruction-induced misinformation or obvious jailbreak-type attacks. We are adding this hypothesis to the revised manuscript.
>
> **v)** We believe there are two concepts getting mixed up. We are not trying to detect correct vs. incorrect answers (in this case, we agree that an LLM can produce incorrect answers with high confidence). Instead, we detect attacked vs. unattacked answers. We support the hypothesis that uncertainty is a good signal for this with Fig. 5, which then led us to verify it with the classifier experiment in Fig. 6.

---

> > ### Author Response · Authors · 2026-03-17
> >
> > **4) Improvement of writing.**
> >
> > **i)** Your interpretation is correct. We have simplified this line in the revised manuscript (see revision).
> >
> > **ii)** Due to the close relation of adversarial attacks and counterfactual examples, namely CEs (see Wachter et al. 2017), we chose to explain the difference in our setup. CEs are defined as the minimal necessary perturbation from the original example in order to shift the outcome (in our case, the answer). With the $\arg\min$ statement, we express this desideratum: finding the minimal distance between the original query $q_i$ and another query $\hat{q}_i$, which would give us $q^*_i$ – the minimally (argmin) perturbed query that leads to a successful attack. We state that we relax this constraint defined by the $\arg\min$, as we do not aim to find the “minimal” counterfactual, but only one that works (i.e., a query perturbation that successfully fulfils the attack).
> >
> > Eq. 2: $a_i$ refers to the original, correct answer (see section 3). $a^*_i$ refers to the answer resulting from a successful attack. However, we have significantly simplified our formalizations, hence this line is not part of the manuscript anymore.
> > Eq. 3: Thank you, we have corrected it.
> >
> > **iii)** Thank you for pointing this out. We agree that $h$ was not adding benefit to the explanation, hence we have also removed it, as our simplification does not call for it anymore.
> >
> > **iv)** Since we added two more models to our evaluations, we reordered the models in the figure to go from highest average performance to lowest. However, performance does not always coincide with model size, which we note as an additional insight.
> >
> > We would greatly appreciate hearing whether we have addressed your concerns and are happy to continue the discussion should any questions remain.

---

> > > ### Author Response · Authors · 2026-03-23
> > >
> > > Dear Reviewer HD7x,
> > >
> > > We appreciate the opportunity to improve the manuscript based on your comments. We believe these updates address the issues raised and would be happy to provide further clarification should any questions remain. Thank you for your time.

---

> > > > ### Comment · Reviewer_HD7x · 2026-03-25
> > > >
> > > > I thank the authors for their detailed and thoughtful responses. I still have two remaining concerns:
> > > >
> > > > (1) Justification of the threat model. If model owners deploy protections on both ends of the communication channel (e.g., using authentication mechanisms such as HMAC), would this effectively eliminate or significantly reduce the feasibility of the proposed threat model? Clarifying this would help better position the practical relevance of the setting.
> > > >
> > > > (2) Rigor of the problem formulation. While the revision simplifies the notation, the new definitions remain insufficiently rigorous. In particular, several of my earlier concerns about the formalization still apply. For example, "in Definition 1, an attack does not need to flip the answer for every query in practice. Moreover, once the output is probabilistic, it becomes unclear how the attack success condition should be formally defined."
> > > >
> > > > Although the high-level intuition behind the proposed methods is understandable, the current formulation does not yet provide a precise or rigorous characterization. I would suggest either: (i) fully formalizing the problem and methods, or (ii) removing the partial formalization and instead presenting the ideas in a more intuitive and informal manner.

---

> > > > > ### Author Response · Authors · 2026-03-27
> > > > >
> > > > > We’re glad to hear that the majority of concerns has been cleared up. Let us explain on the remaining two points:
> > > > >
> > > > > 1. **Threat model:** It is true that in an end-to-end encrypted environment, where both input and output channels are fully authenticated, the feasibility of an external MitM attack would be reduced. However, as observed in other papers targeting cybersecurity topics, the function of a threat model is not to cover all possible scenarios, but to specify a specific, realistic setup in which the study operates. In our case, we chose to explore the attack surface in an untrusted client-side setting. This remains a highly realistic scenario due to the widespread use of third-party API intermediaries and browser extensions, which can modify content before any encryption or signing occurs.
> > > > >
> > > > > 2. **Definitions:**
> > > > >
> > > > >  (i) We clarify that Def. 1 is formulated as an *instance-level* attack. It defines the necessary and sufficient conditions for a single perturbed query $\hat{q}_i$ to be considered a successful attack. It is not intended to define the “global success rate” of a series of attacks. We will clarify this in the manuscript.
> > > > >
> > > > > (ii) We appreciate this observation and agree that LLM generation is inherently non-deterministic (for temperatures $> 0$). To address this, we had added a footnote in Section 3 of the revision clarifying our handling of multiple correct answers. We believe the current point raised is related to this, and would like to clarify that this stochasticity does not conflict with Def. 1. Our formulation treats the generated output $g(\hat{q}_i)$ as a realized instance which is then evaluated by the oracle $\Phi$. While the specific output may vary across trials, the success condition remains the same: whether that specific realization is correct or incorrect. We will explicitly discuss this probabilistic behavior in the final manuscript, but we would welcome any further guidance if you feel this non-determinism necessitates a more fundamental change to the definition itself.
> > > > >
> > > > > Since we have had mixed responses from reviewers regarding the level of formalization in the paper (either to simplify, to remove, or extend further), we suggest the following: we will keep the main text of the manuscript simplified in order to ease the read. However, we will outsource the expanded formalization into the appendix for the interested reader. We hope this can be a suitable compromise.
> > > > >
> > > > > Once again, we highly appreciate your feedback on our work, and are looking forward to hear your response.

---

### Review · Reviewer_XLc6 · 2026-02-27

**Summary Of Contributions:**

The paper introduces Xmera, a framework for evaluating man-in-the-middle (MitM) attacks on LLM factual recall in closed-book QA settings. Three attack types are proposed: (1) alpha-xmera, which appends misleading instructions to the original query (e.g., "respond with a wrong answer"); (2) beta-xmera, which prepends factually incorrect context by swapping entities in a true fact; and (3) gamma-xmera, which injects semantically unrelated but well-formed noise. The authors evaluate these attacks across several LLMs and find that the simplest attack (alpha-xmera) is the most effective, achieving up to ~85% attack success. They also train Random Forest classifiers on uncertainty features to detect attacked queries, reporting AUC up to ~96%. The paper claims five contributions, including the framework, the attacks, a robustness audit, the detection classifiers, and a dataset release.

Key strengths:
- The paper addresses a real concern about the integrity of LLM-generated answers when inputs can be tampered with.
- The empirical finding that trivial instruction-based attacks are highly effective is useful and worth documenting.
- The experiments are reasonably thorough, covering multiple models and QA datasets.

Key weaknesses:
- The threat model is not convincingly justified. If an attacker has MitM access to the communication channel between user and LLM, it is unclear why they would limit themselves to modifying the query rather than simply rewriting the response, serving a fake endpoint, or exfiltrating data. The paper needs to argue why query-only modification represents a realistic and important attack surface, and it does not do this adequately.
- The formalization is over-engineered relative to what is actually being done. The mathematical definitions (e.g., Definition 2 with indicator functions and argmax notation) dress up what are fairly simple operations (appending instructions, prepending false context, injecting noise) without providing proportional insight.
- The beta-xmera attack is confusingly presented. In Figure 2, it appears that the modification targets the answer rather than the question. What is actually happening is that a false context sentence (with a swapped entity) is prepended to the unchanged question, but the paper's language and notation blur the distinction between "modifying the query" and "prepending misleading context to an unchanged query." These are conceptually different operations, and the latter is closer to a simplified RAG poisoning scenario than a traditional MitM attack.
- The defense evaluation has a methodological flaw: there is no adaptive attacker. The classifier is trained on uncertainty signatures of these specific three attacks and tested on the same distribution. An attacker aware of the defense could craft attacks that maintain low uncertainty, and the paper does not address this. The ~96% AUC is therefore not evidence of a robust or generalizable defense, despite the paper's framing of it as "a first checkpoint toward user cyberspace safety."
- Five claimed contributions is excessive. The dataset release and classifier are minor supporting work. The real contributions are the framework/attacks and the uncertainty analysis.

**Audience:**

Yes

**Audience Explanation:**

The question of how LLM outputs can be corrupted through input manipulation is timely and relevant to TMLR's audience. The simplicity of the findings, the issues with the defense proposed, and the lack of motivation for the threat model cut against this. I marked it as interesting, although it's a fairly weak yes.

**Claims And Evidence:**

No

**Claims Explanation:**

The empirical results themselves (attack success rates, uncertainty measurements) appear to be correctly computed, but several of the paper's higher-level claims are not convincingly supported.

First, the threat model lacks justification. The paper frames the attack as a MitM scenario where an intermediary modifies user queries before they reach the LLM. But it does not explain why an attacker with this level of access would restrict themselves to query modification. An attacker who controls a proxy layer or browser extension could simply rewrite the LLM's response directly, which would be far more effective and reliable. The paper needs to make a clear argument for why query-only perturbation is a realistic and interesting attack surface, separate from response tampering. Without this, the entire threat model feels under-motivated.

Second, the defense claim is overstated. The ~96% AUC for detecting attacks via Random Forest classifiers on uncertainty features sounds impressive, but this is a classifier trained and tested on the specific uncertainty signatures of three known attacks. There is no evaluation against an adaptive attacker who is aware of the detection mechanism and could design attacks to evade it. This is a methodological flaw: in adversarial settings, evaluating a defense only against the attacks it was trained on tells you very little about its robustness. The paper's framing of this as a practical safety mechanism ("first checkpoint toward user cyberspace safety") is not supported by the evidence presented.

Third, the formalization does not clearly support the claims of a "principled" or "theory-grounded" framework. The mathematical definitions formalize simple operations (appending text, prepending false context, injecting noise) with heavy notation, but this formalism does not generate new insights or predictions beyond what the plain English descriptions already convey. The gap between the formal framework and the actual implementation is large.

**Requested Changes:**

Critical changes (required for acceptance):

1. Justify the threat model. The paper must clearly explain why an attacker with MitM access would be limited to modifying the query rather than the response. Section 4.1 mentions "proxy gateways, browser extensions, or proxy layers" but does not argue why an attacker in these positions would only modify the input side. Without this justification, the core premise of the paper is under-motivated. If the authors believe there are realistic scenarios where query modification is possible but response modification is not, they need to describe those scenarios concretely.

2. Evaluate the defense against an adaptive attacker. The current defense evaluation only tests against the specific attacks the classifier was trained on. This is a methodological flaw in any adversarial evaluation. The authors should at minimum discuss what an adaptive attacker could do to evade the uncertainty-based detection (e.g., crafting attacks that maintain low uncertainty), and ideally evaluate against such attacks. Without this, the ~96% AUC number should not be presented as evidence of a robust defense.

3. Reduce the formalization or clarify it's contribution to the paper. The mathematical framework (Definitions 1-3) adds notation without generating insights beyond plain descriptions. Either simplify the presentation or show how the formalism enables analysis that would not be possible otherwise.

Suggested improvements (not required for acceptance):

5. The paper claims five contributions, which overstates the novelty. The dataset release and the Random Forest classifiers are supporting work, not standalone contributions. Consider consolidating.

6. The related work section covers relevant areas but does not clearly articulate how this paper sits at the intersection of factual knowledge, counterfactual robustness, prompt attacks, and misinformation research. Strengthening the connective narrative would help the reader understand the paper's positioning.

7. The models evaluated (LLaMA-2-13B, Mistral-7B) are somewhat dated. Including results on more recent model families would strengthen the empirical contribution.

8. The answer-matching evaluation (checking if the ground truth is a substring of model output) is coarse and could produce noisy results from partial entity matches. Consider whether a more precise evaluation metric is warranted.

---

> ### Author Response · Authors · 2026-03-17
>
> Thank you for your thorough review. We truly appreciate the time you have taken and the constructive feedback. In the following, we address the raised concerns one by one. *Please also refer to the uploaded revised manuscript, which includes the changes marked in blue and deletions marked in red.*
>
> **1. Threat model justification:** Thank you for raising this point - we believe the manuscript would indeed benefit from the following clarification. We focus on query-level perturbation due to the asymmetric integrity protection prevalent in enterprise LLM deployments. While a MitM adversary may be able to access traffic in both directions, modern API gateways typically implement Unidirectional Response Signing (e.g. via HMAC) to ensure output integrity. In this architecture, the gateway verifies that the response originates from a trusted LLM IP address and matches a specific cryptographic signature; any downstream modification by an attacker would invalidate this signature and trigger a security rejection.
> Conversely, the “upstream” query path is treated as an untrusted ”write” surface. Because queries originate from diverse, variable user IPs, gateways rarely enforce rigid identity or integrity checks on the input. By poisoning the query before it reaches the server, the adversary exploits this lack of input verification to force the LLM to generate a signed, authentic response containing the intended falsehood. Thus, we believe query modification is the primary viable path for a stealthy attack that survives both cryptographic and identity-based filters. From a scalability and impact perspective, we believe that a query-side attack is more parsimonious and more scalable as the attacker could exploit specific query types on certain topics potentially affecting millions of users at the same time.
>
> **2. Extensive formalization:** Thank you for addressing this. Indeed, after careful consideration, we agree with you that the formalization is too extensive and is adding more confusion than benefit. Hence, we have significantly simplified Sections 3 and 4. Please refer to the attachment for the exact changes.
>
> **3. Confusion about beta-Xmera attack:** Figure 2 states that an attack $\chi_\beta$ is made up of the perturbed fact ($\phi(f_i)$ plus the original question $q_i$. We find it a bit difficult to see how the modification could mistakenly be attributed to the answer instead of the question, and we would appreciate further clarification on how we can make this less confusing.
>
> **4. No evaluation on adaptive attacker:** We indeed did not additionally evaluate in scenarios where the adversary might be capable of more elaborate attacks. Hence, we explicitly acknowledge in section 6 that *“further research should explore other possible defense signals beyond uncertainty”*. In this paper, we argue that uncertainty can be an early indicator of the presence of a malicious actor – how cleverly an adversary can engineer an attack is beyond the scope of our paper.
>
> **5. Too many claimed contributions:** We are happy to rethink the presentation of our contributions. In particular, we believe that the key points are the formalization of MitM attacks in LLMs, the attack evaluation in the context of factual memory, and the attack alert based on uncertainty, as we find this could be interesting for future researchers building detection mechanisms upon this. Please see the attached manuscript for the rewritten section.
>
> **How are we addressing the critical changes required for acceptance:**
>
> **1.** We have added a more elaborate justification for our threat model (see above points).
>
> **2.** As requested, we have picked up the possibility of advanced attackers when discussing future work, and are using it as additional motivation for why more diverse defense signals can be of interest in papers building upon our findings. We present the results of our classifier as robust against Xmera-type attacks and will refine our formulations where needed to express this explicitly.
>
> **3.** We have significantly simplified the formalizations in Sections 3 and 4.

---

> ### Author Response · Authors · 2026-03-17
>
> **Suggested improvements:**
>
> **5.** We have adjusted our list of contributions as stated above.
>
> **6.** Our work sits at the intersection of these domains by utilizing prompt-based attacks as the adversarial vector to inject misinformation, while drawing inspiration from counterfactual explainability to formalize the construction of these perturbations as principled interventions. We have added this clarification to our related work section.
>
> **7.** Thank you for this suggestion. We added more recent models, LLaMA-3.1-8B and Mistral-Nemo-12B, to our experiments. The results are added to the revised manuscript.
>
> **8.** While more complex metrics exist, substring matching is appropriate here as our ground-truth answers consist of short, specific entities (1-2 words). The likelihood of these unique sequences appearing in the model's output by chance or in a semantically incorrect context is negligible. Furthermore, because this metric is applied consistently across all experimental groups, it provides a robust basis for comparing the relative impact of our attacks, even if absolute scores contain minor noise.
>
> We would greatly appreciate hearing whether we have addressed your concerns and are happy to continue the discussion should any questions remain.

---

> > ### Comment · Reviewer_XLc6 · 2026-03-17
> > **Pretty happy with the changes, a few remaining questions.**
> >
> > Overall, I thank the authors for their careful consideration of my comments. I think this improves the paper and mostly addresses my comments.
> >
> > I think the most important part is the justification of the threat model. However, one question still remains for me: why not simply replace the query with an alternative query entirely? It seems like you could essentially choose the LLM response by overwriting the query with "repeat the following: <desired attacker response>" and that this would likely avoid the uncertainty detection method proposed.
> >
> > Re: Figure 2, maybe it would be helpful to include a plaintext description of the actual query? I recognize that this is implied symbolically, but it's something that I missed on my first reading.

---

> > > ### Author Response · Authors · 2026-03-18
> > >
> > > Thank you for your very quick response. We’re glad that we could clear up most concerns.
> > >
> > > **Regarding the threat model:** yes, technically that scenario is also possible (assuming an attacker can modify the entire query rather than only adding text on top). However, we believe it comes down to the *semantics* of the LLM’s answer and the conversational flow the user expects.
> > > If an attacker forces the model to just say the wrong answer and nothing else, it would likely look very odd or “robotic” to the user. If the attacker tries to craft a more elaborate fake response, it might not match the model’s usual tone, formatting, or style. By adding adversarial cues to the query instead, we trick the model into producing the wrong response naturally and contextually plausible. This makes the output feel much more trustworthy to the user and significantly reduces the chance of them noticing any obvious tampering. How far an attacker can go in terms of such non-obvious manipulations is an excellent question for future research.
> > >
> > >  **Regarding Fig. 2:** That makes sense. We will add this into the final manuscript, thank you for clarifying.
> > >
> > > We hope that we have addressed all concerns in the category *“Are the claims made in the submission supported by accurate, convincing and clear evidence?”*, and would be happy to have your support of our paper. If there are any more questions, we’re of course available to discuss further.

---

> > > > ### Comment · Reviewer_XLc6 · 2026-03-18
> > > >
> > > > That makes sense, thank you. Can you please update your manuscript with a discussion that addresses this? I think it is important to motivate the specific work you've done.

---

> > > > > ### Author Response · Authors · 2026-03-18
> > > > >
> > > > > We have reuploaded a revised version of the manuscript. We added the above clarification about the threat model, as well as the exact resulting queries in all three attacks (Figures 1, 2 and 3). We hope that they appear clearer to the reader this way.

---

### Review · Reviewer_Rznm · 2026-03-05

**Summary Of Contributions:**

This paper studies adversarial prompt manipulation under a man-in-the-middle (MitM) threat model, where an intermediary modifies user queries before they reach an LLM. The authors propose the Xmera framework, which defines three attack variants that perturb the input query through instructions, false contextual facts, or unrelated context to induce incorrect answers. Experiments on closed-book QA benchmarks show that such perturbations can significantly degrade factual accuracy, and the authors explore uncertainty signals as a possible detection mechanism.

**Audience:**

Yes

**Audience Explanation:**

Understanding how prompt manipulation affects factual recall is relevant for research on LLM security, robustness, and reliability. However, the empirical findings, namely that simple prompt manipulations can induce factual errors, may already be expected given existing literature on prompt injection and jailbreak attacks.

**Claims And Evidence:**

No

**Claims Explanation:**

The experiments show that prompt perturbations can degrade factual QA accuracy, but the novelty of the proposed attacks is unclear. The $\alpha$-Xmera attack essentially appends an instruction such as “Respond with a wrong answer,” which closely resembles standard instruction-based jailbreak prompts. Similarly, the fact-aware attacks resemble context-based or few-shot jailbreak strategies rather than fundamentally new attack mechanisms.

In addition, the paper attributes robustness differences primarily to model size, but the evaluated models come from different model families and training corpora, making such conclusions difficult to justify.

**Requested Changes:**

- Clarify how Xmera attacks differ from standard instruction-based jailbreak or prompt-injection techniques, as the current formulations appear very similar.

- Provide stronger baseline comparisons with existing jailbreak or prompt-injection methods.

- Improve the positioning with respect to prior work on LLM knowledge probing and factual robustness (e.g., recent work studying model knowledge status and behavior with and without context [1]).

- Avoid attributing robustness differences solely to model size, since the evaluated models differ in model family and training data.

[1] KScope: A Framework for Characterizing the Knowledge Status of Language Models. NeurIPS 2025.

---

> ### Author Response · Authors · 2026-03-17
>
> Thank you for taking the time to review our paper and for pointing out ways to improve. We have revised the manuscript to address your concerns. In the following, we provide answers to the specific points you raised. *Please also refer to the uploaded revised manuscript, which includes the changes marked in blue, and deletions marked in red.*
>
> **1) Attack novelty and comparison to other attacks:** Our primary goal is not to introduce entirely new attack mechanisms, but rather to provide a systematic evaluation of how these fundamental attack vectors manifest specifically within factual QA settings. While the underlying mechanics (e.g., instruction- or context-based) share similarities with general jailbreaks, their impact on factual integrity in QA is a distinct area of concern that warrants this focused analysis. This approach aligns with TMLR’s core criteria, which emphasize the scientific soundness and empirical support of claims over the novelty of the mechanism itself. Ultimately, we believe our systematic study provides valuable, sound insights for the community regarding the fragility of factual retrieval in particular.
>
> **2) Baseline comparisons:** We are pointing to jailbreak and prompt-injection methods in our related work section. However, as explained above, *our goal is not to outperform any existing methods*. Our aim is to assess attacks aimed at manipulating factual QA scenarios and to show how easily implementable they would be for an attacker.
>
> **3) Improve positioning w.r.t prior work:** We have extended our related work section to address our position with respect to works on internal LLM knowledge, making our adversarial perspective clearer. Please see the uploaded revision.
>
> **4) Attributing robustness to model size:** Thank you for pointing this out. We have run additional experiments with models from the same family, adding LLaMA-3-8B and Mistral-Nemo-12B. Indeed, we have found that model size is one factor, but not the only one, as newer models have outperformed their older counterparts, even though they are not always larger (e.g., in the case of the two LLaMA models). Please refer to the attached manuscript for the added results in uncertainty evaluation and attack detection.
>
> We hope we have addressed your concerns adequately and are happy to answer any additional questions.

---

> > ### Author Response · Authors · 2026-03-23
> >
> > Dear Reviewer Rznm,
> >
> > We appreciate the opportunity to improve the manuscript based on your comments. We believe these updates address the issues raised and would be happy to provide further clarification should any questions remain. Thank you for your time.

---

### Decision · Action_Editor_ZCvQ · 2026-04-06

**Recommendation:** Reject

**Audience:**

Yes

**Audience Explanation:**

LLM security remains an important and timely topic.  The general subject the paper tackles is interesting.

**Claims And Evidence:**

No

**Claims Explanation:**

All reviewers expressed concerns with the threat model.  Furthermore, despite the rebuttal discussion, the majority of reviewers voted in favor of rejection.  Additional concerns include the theoretical framework introduced to discuss the attacks, metrics used, and the relation to prior work.

The work has promise, and I recommend a major revision to address the reviewers' concerns and overall feedback.  Additionally, the following suggestions would help improve the paper:
- perform the following: "we will keep the main text of the manuscript simplified in order to ease the read. However, we will outsource the expanded formalization into the appendix for the interested reader."  At present, the theoretical framework does not serve any theoretical proofs, so they overcomplicate the paper and greatly reduce overall clarity
- streamline the writing and focus on clarity
- situate the paper's contributions given existing published papers.  In particular, MitM for LLMs are not new [1,2], and discussing the contributions in the context of existing work will help ground the attacks
- add missing citations and examples, e.g.,: "API gateways often use cryptographic signatures (e.g., HMAC) to verify that
responses originate from a trusted LLM IP," " the “upstream” query path is typically an unsigned, untrusted “write” surface," "Users often rely on black-box systems for factual information, with no visibility into how their inputs are handled internally. Even when encrypted channels are used, tampering can occur upstream or downstream from the model itself."

# References
[1] He, Pengfei, et al. "Red-teaming llm multi-agent systems via communication attacks." Findings of the Association for Computational Linguistics: ACL 2025. 2025.

[2] Shaikh, Asif, Aygun Varol, and Johanna Virkki. "From Prompts to Motors: Man-in-the-Middle Attacks on LLM-Enabled Vacuum Robots." IEEE Access (2025).

[3] Yan, Bingyu, et al. "Attack the messages, not the agents: A multi-round adaptive stealthy tampering framework for llm-mas." Proceedings of the AAAI Conference on Artificial Intelligence. Vol. 40. No. 35. 2026.

**Resubmission Of Major Revision:**

The authors may consider submitting a major revision at a later time.